# Biodegradable Stents: A Breakthrough in the Management of Complex Biliary Tract Injuries: A Case Report

**DOI:** 10.3390/reports7040095

**Published:** 2024-11-09

**Authors:** Ottavia Cicerone, Giulio Di Gioia, Maria Pajola, Anna Gallotti, Antonio Mauro D’Agostino, Nicola Cionfoli, Riccardo Corti, Pietro Quaretti, Marcello Maestri

**Affiliations:** 1Dipartimento di Scienze Clinico-Chirurgiche, Diagnostiche e Pediatiche, University of Pavia, 27100 Pavia, Italy; ottavia.cicerone01@universitadipavia.it (O.C.); giulio.digioia01@universitadipavia.it (G.D.G.); maria.pajola01@universitadipavia.it (M.P.); 2Radiologia, Fondazione IRCCS Policlinico San Matteo, 27100 Pavia, Italy; a.gallotti@smatteo.pv.it; 3Radiologia Interventistica, Fondazione IRCCS Policlinico San Matteo, 27100 Pavia, Italy; a.dagostino@smatteo.pv.it (A.M.D.); n.cionfoli@smatteo.pv.it (N.C.); ri.corti@smatteo.pv.it (R.C.); p.quaretti@smatteo.pv.it (P.Q.); 4Chirurgia Generale I, Fondazione IRCCS Policlinico San Matteo, 27100 Pavia, Italy

**Keywords:** biliary tract injury, biodegradable stent, postoperative complication, case report

## Abstract

**Background and Clinical Significance**: Biliary tract injuries are a recognized complication of laparoscopic cholecystectomy. Early diagnosis and prompt management are crucial to minimize complications such as bile leaks, strictures, and fistula formation. This case report highlights the use of a biodegradable biliary stent in managing a complex biliary injury and discusses the impact of delayed diagnosis on treatment outcomes. **Case Presentation**: We present the case of a 30-year-old male who sustained a Strasberg E2 biliary tract injury during a laparoscopic cholecystectomy. Initially misdiagnosed, the injury was only recognized on the fourth postoperative day. The patient underwent a Roux-en-Y hepaticojejunostomy and subsequently developed a postoperative biliary fistula, which was managed with percutaneous drainage. A biodegradable biliary stent was later placed to address a stricture and minimize the need for future interventions. One year later, the patient presented with symptoms of cholangitis, and radiological findings revealed a narrowing of the biliary lumen. The stricture was resolved and an endoscopic gastrojejunal shunt was placed to prevent further complications. The patient is currently in good condition with no signs of further complications. **Conclusions**: This case emphasizes the importance of early diagnosis in managing biliary tract injuries and highlights the potential of biodegradable stents to reduce the need for repeat interventions. Despite a delayed diagnosis necessitating complex surgical procedures, the use of a biodegradable stent proved effective in managing postoperative complications. Further studies are needed to evaluate the long-term efficacy of biodegradable stents in similar clinical scenarios.

## 1. Introduction and Clinical Significance

With the advent of laparoscopic cholecystectomy, there has been an increased incidence of biliary tract injuries. The main causes of these injuries include the incorrect interpretation of biliary anatomy, chronic inflammation in the Calot triangle, anatomical anomalies such as Mirizzi syndrome, and surgeon inexperience [1]. CT scans and magnetic resonance cholangiopancreatography (MRCP) are recommended for identifying biliary tract injuries and assessing any associated vascular damage [2]. The most used classification for these injuries is the Strasberg system: Strasberg A, B, C if the main biliary duct is not involved, and D if there is a partial section of the main biliary duct. Type E follows the Bismuth classification for extrahepatic biliary tract injuries (Table 1) [1,3].

Management by a multidisciplinary team consisting of a hepatobiliary surgeon, interventional radiologist, and endoscopist has been shown to have a prognostically positive impact, whereas inexperience in this area appears to be an independent risk factor [1,4]. For type A injuries, an endoscopic or percutaneous approach is indicated, while types C and D may require surgical or percutaneous treatment depending on the case; types B and E require surgical intervention. For type E, the treatment of choice is surgical intervention with a Roux-en-Y hepaticojejunostomy [2]. Reconstruction of biliary continuity is complex and may be complicated by the onset of fistulas, strictures, and dehiscence: in such situations, the intervention of an interventional radiologist with the placement of a percutaneous internal–external hepatic biliary drainage (IEPTBD) is crucial to salvage the anastomosis [5].

In extreme cases, patients may develop progressive hepatic insufficiency necessitating liver transplantation [6].

It is crucial to note that community hospitals may lack the specialized resources and expertise necessary for managing complex biliary tract injuries. Therefore, whenever possible, these patients should be referred to dedicated Hepato-Pancreato-Biliary (HPB) centers, where a multidisciplinary team can ensure optimal management [7].

Clinical significance: This case report presents a novel approach to managing complex biliary tract injuries using biodegradable stents, demonstrating the potential advantages of this technology in reducing complications and reinterventions

## 2. Case Presentation

In our case study, we present a 30-year-old man with no significant past medical history referred to our center following a major biliary tract injury (Strasberg E2) due to a laparoscopic cholecystectomy performed at another center. Initially, the injury was misdiagnosed as a minor bile leak, leading to the critical injury being missed for several days. It was only recognized on the fourth postoperative day through an MRI (Figure 1). The patient was then transferred to our HPB center in septic shock and with hemoperitoneum. An ERCP was performed to visualize the injury. The patient urgently underwent surgery. At the laparotomy, no vascular injury was identified, and the hemoperitoneum was attributed to the transection of bile duct vessels. A Roux-en-Y bilioenteric anastomosis was performed, and two abdominal drains were placed. The decision to perform a Roux-en-Y hepaticojejunostomy was made after thorough multidisciplinary evaluation, weighing the risks of delaying the intervention against the necessity for definitive management of the biliary injury. On the third postoperative day, the patient developed a fistula near the surgical suture, suspected clinically and by the increase in bile-like fluid in the drains; this was subsequently confirmed by appropriate imaging studies. After a multidisciplinary evaluation, a percutaneous internal–external hepatic biliary drainage (IEPTBD) was indicated by the interventional radiologist to protect the anastomosis and avoid a second surgical intervention. A close follow-up was initiated during which the fistula resolved. Given the complexity of the biliary injury, a biodegradable stent was chosen for its potential to reduce the need for future interventions and minimize the complications typically associated with traditional stents. The biliary stricture was treated by the interventional radiologist through forced dilation, followed by the insertion of the biodegradable stent. One month later, the percutaneous drainage was successfully removed. The detailed series of treatments is comprehensively depicted in Figure 2.

### 2.1. Follow up

The follow-up protocol included MRCP at regular intervals of 3 and 6 months, which confirmed the successful resolution of the biliary injury and the patency of the duct.

After one year, the patient presented to the emergency department with elevated bilirubin and gamma-glutamyl transferase (gamma-GT) levels, along with a fever, consistent with a diagnosis of cholangitis. Radiological investigations revealed a narrowing of the biliary lumen, despite it remaining patent. To prevent future complications, monitor the condition, and intervene if necessary while avoiding repeated percutaneous transhepatic cholangiographies (PTCs), a decision was made to place a gastrojejunostomy (GJ) shunt. It was implemented using an endoscopic technique that creates a bypass between the stomach and jejunum [8]. The patient is currently in good condition, with normal laboratory tests and no signs of further complications. However, there remains a persistent risk of strictures in the future.

### 2.2. Patient’s Perspective

The patient expressed gratitude for the opportunity to undergo a procedure that allowed him to resume a normal life without the frequent need for stent changes or regular visits to interventional radiologists. He particularly appreciated the idea of a treatment option that offered long-term relief and minimized the need for invasive interventions. The patient also acknowledged the seriousness of his condition and expressed relief at having survived a life-threatening complication. Today, he has fully resumed his work and daily activities and reports a significant improvement in his quality of life.

The patient’s perspective documented here is based on direct feedback during follow-up consultations, reflecting his personal experiences and sentiments.

## 3. Discussion

Recent studies have shown that this type of biodegradable stent is not inferior to traditional plastic stents in maintaining biliary duct patency and has significantly reduced the number of reinterventions required for replacement and removal, thereby lowering overall treatment costs [9]. Compared with plastic and metallic stents, biodegradable stents provide the added benefit of self-degradation, which eliminates the need for additional procedures to remove the stent, thus lowering the burden on both patients and healthcare systems. However, there is a slightly higher risk of cholangitis with biodegradable stents compared with plastic stents. This risk is thought to be associated with the degradation process, as the breakdown of the stent material may trigger local inflammatory reactions. Nevertheless, further research is required to understand the exact mechanisms and to optimize the biodegradability of the materials used [9]. The most commonly used biodegradable materials are synthetic polymers such as polydioxanone, polylactic acid, and polycaprolactone (used, for example, in several commercially available sutures) [10,11] (Table 2).

In our study, we used a stent made from polydioxanone due to its biocompatibility and ability to maintain mechanical integrity over a prolonged period.

Research in this area has shown promising results despite being relatively unexplored and necessitating further studies. It is anticipated that the adoption of new technologies, such as innovative materials, manufacturing techniques, and drug-delivery technologies, will facilitate the development of new biodegradable stents.

The repair of iatrogenic biliary tract injuries is a complex field characterized by a high risk of complications and significant mortality. Accurate identification and classification of the injury, as well as management at a specialized HPB center, are crucial for improved outcomes. In this context, proper planning and individualized treatment have been shown to have a greater prognostic impact than the timing of the surgical intervention [12]. Furthermore, the development of fistulas and stenosis is not uncommon in these procedures, making the collaboration with interventional radiologists essential for effective postoperative management.

As the technology of biodegradable stents continues to evolve, larger-scale studies are necessary to fully evaluate their long-term efficacy and safety, and to identify the patient populations that may benefit most from their use.

It is important to note that in this case, the diagnostic delay likely played a significant role in complicating the patient’s postoperative course. Early identification of biliary tract injuries is crucial to reduce the risk of bile leaks, strictures, and fistula formation. In this patient, the delay in recognizing the injury until the fourth postoperative day necessitated more complex interventions, including a Roux-en-Y hepaticojejunostomy and the subsequent placement of a biodegradable stent. Additionally, early intervention could have reduced the likelihood of biliary fistula development, which required prolonged percutaneous drainage and increased the overall duration of the patient’s recovery. This emphasizes the importance of prompt diagnosis and highlights how delays can lead to more challenging treatment pathways and extended recovery times.

### Limitations

Despite the successful use of biodegradable stents in this case, certain limitations should be noted. A slightly higher risk of cholangitis has been reported in the literature and was also observed in our patient. Additionally, longer follow-up periods are necessary to fully assess the long-term efficacy and safety of this intervention. Further research is required to better understand long-term outcomes and to identify the optimal patient population for this approach.

## 4. Conclusions

The aim of this article is to highlight how the use of biodegradable biliary stents holds promise in managing patients with complex biliary tract injuries. Their functional equivalency to current plastic or metallic stents, combined with fewer required procedures, makes them an appealing alternative in managing the well-known complications of postoperative care in such interventions. Their ability to reduce the frequency of reinterventions, along with their favorable biocompatibility, positions biodegradable stents as a highly promising option in the surgical management of complex biliary tract injuries.

## Figures and Tables

**Figure 1 reports-07-00095-f001:**
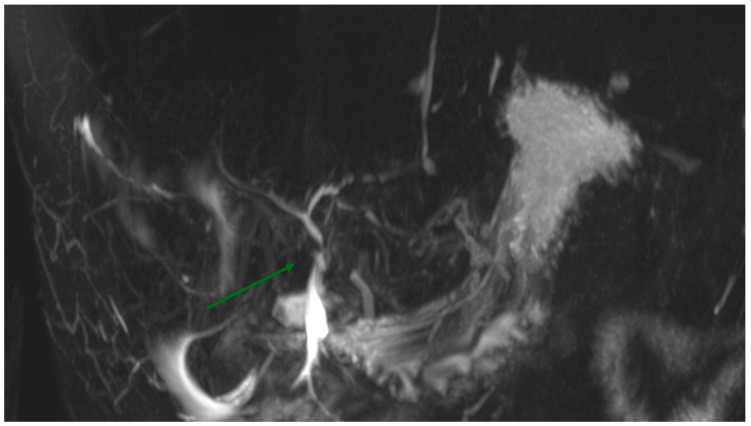
MRI showing the biliary tract injury. The arrow points to the Strasberg E2 biliary tract injury.

**Figure 2 reports-07-00095-f002:**
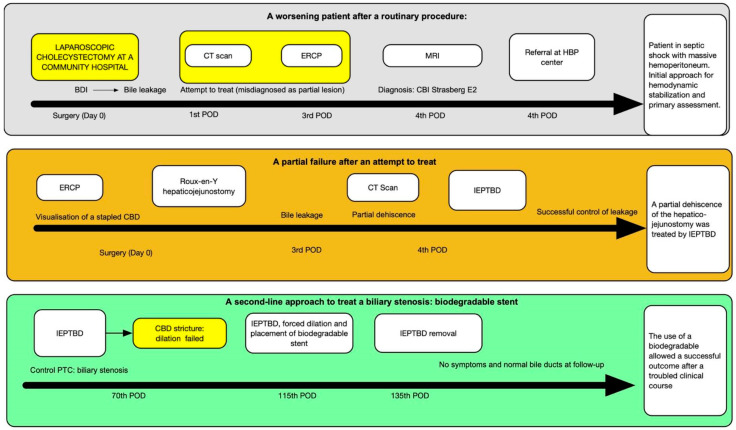
Flowchart depicting the case-specific diagnostic and therapeutic strategy.

**Table 1 reports-07-00095-t001:** Classification of biliary tract injuries.

Type	Injury Description	
A	Cystic duct injury or leaks from minor ducts in liver bed	
B	Occlusion of aberrant right hepatic duct	
C	Transection of aberrant right hepatic duct	
D	Lateral injury to a major bile duct	
E (Bismuth transection of >50%)	Injury to the main hepatic duct; classified according to level injury:	
	E1	More than 2 cm from biliary confluence
	E2	Less than 2 cm from biliary confluence
	E3	Injury at the confluence, that is intact
	E4	Transection of the biliary confluence
	E5	Type C with concomitant main hepatic duct injury

**Table 2 reports-07-00095-t002:** Biodegradable stent materials and their properties.

Material	Degradation Product	Degradation Time in Bile	Mechanical Properties Retention Time in Bile
PLGA (80/20)	Lactic acid, glycolic acid	2–3 weeks	4 days
PGA	Glycolic acid	2 months	1 week
PDX	Glyoxylic acid, glycine	3–5 months	3 months
PLLA	Lactic acid	>9 months	>8 weeks
PCL	Hydroxycaproic acid	>70 days	/
WE43 (Mg)	Mg(H_2_PO_4_)_2_	/	>60 days

PLGA: poly l-lactide-co-glycolide; PGA: poly(glyoxylic acid); PDX: polydioxanone; PLLA: poly(L-lactic acid); PCL: polycaprolactone; Mg: magnesium.

## Data Availability

The original data presented in this study are available on reasonable request from the corresponding author. The data are not publicly available due to privacy concerns.

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
