# Peer review of "Biodegradable Stents: A Breakthrough in the Management of Complex Biliary Tract Injuries: A Case Report"

_reports, 2024, doi:10.3390/reports7040095_

Round 1

Reviewer 1 Report

Comments and Suggestions for Authors

I read with interest the BDI manuscript and i have three comments for authors to respond: (a) a community hospital made an attempt to repair the BDI - is this acceptable for a community hospital to NOT refer HPB centre but attempt to repair the injury? Discuss this in the manuscript. (b) there is no mention of bleeding complication, but at the HPB centre, patient had septic shock and hemoperitoneum both. So was there any vascular injury? What was the cause of hemoperitoneum? (c) The patient was septic and unwell with hemoperitoneum. How can one decide to do HJ in such a sick patient? Is this something that is reasonable or unreasonable? 

I wonder what is your view on future stricture risk of this patient? DO you think he or she is fine or there is a life long risk with relapse a matter of time?

What do you mean by GJ shunt? You mean to state gastrojejunostomy anastomosis? Please explain how this helps reduce risk of biliary sepsis. I am unclear. 

Patient perspective has to be endorsed and certified by patient. Please confirm if this is your perspective of what your patient ought to feel, or is this truly the patient perspective as per what patient has typed or told to you to type?

Author Response

Comment 1: a community hospital made an attempt to repair the BDI - is this acceptable for a community hospital to NOT refer HPB centre but attempt to repair the injury? Discuss this in the manuscript.

Response 1: This discussion has been included at the end of the Introduction section, as per the reviewer's comment, and it is highlighted in the text. I have also included a reference.

Comment 2there is no mention of bleeding complication, but at the HPB centre, patient had septic shock and hemoperitoneum both. So was there any vascular injury? What was the cause of hemoperitoneum?

Response 2: I have provided a more detailed explanation in the Case Presentation section. These points are highlighted in the text.

Comment 3: The patient was septic and unwell with hemoperitoneum. How can one decide to do HJ in such a sick patient? Is this something that is reasonable or unreasonable? 

Response 3: I have provided a more accurate description of the patient's presentation at our center in the Case Presentation section. It is highlighted in the text.

Comment 4: I wonder what is your view on future stricture risk of this patient? DO you think he or she is fine or there is a life long risk with relapse a matter of time?

Response 4: The potential future risk of strictures is discussed at the end of the Follow-Up section and it is highlighted in the text.

Comment 5: What do you mean by GJ shunt? You mean to state gastrojejunostomy anastomosis? Please explain how this helps reduce risk of biliary sepsis. I am unclear.

Response 5: A more precise explanation is provided in the Follow-Up section, and I have also included a reference regarding the technique. 

Comment 6: Patient perspective has to be endorsed and certified by patient. Please confirm if this is your perspective of what your patient ought to feel, or is this truly the patient perspective as per what patient has typed or told to you to type?

Response 6: I have provided a more detailed explanation in the patient's perspective section.

Reviewer 2 Report

Comments and Suggestions for Authors

Comments to the Authors

The manuscript from Ottavia Cicerone et al. entitled “Biodegradable Stents: A Breakthrough in the Management of Complex Biliary Tract Injuries: A Case Report” (Manuscript ID: reports-3265946) seems interesting. There are points which need to be addressed.

1.     Page 2, “Initially misdiagnosed.” What kind of misdiagnosis was it? What disease was diagnosed? Was the existence of the disease missed?

2.       “HPB” should be fully spelled in their first appearance in the main text.

3.       Figure 1, authors should describe the imaging findings of the MRI scan in the legend. Also, authors can indicate where the problem, biliary injury, is with arrows in the image for readers to understand.

4.       Figure 2, the words are too small to see. Please create a better “Flowchart” for readers.

5.       “gamma-GT” should be describe in fully-spelled.

6.       In Discussion, “Recent studies have shown that this type of biodegradable stent is not inferior to traditional plastic stents in maintaining biliary duct patency and has significantly reduced the number of reinterventions required for replacement and removal, thereby lowering overall treatment costs.” The reference #7 (Eur J Radiol. 2020;125:108899) should be inserted in the end of the first sentence where the content is described.

7.       Table 2, which one was used for this patient? Why and how did authors choose the one? Please describe in Case Presentation or Discussion.

8.       In Conclusions, “in managing these patients.” Readers may not understand what  “these patients” are. Authors should describe here again what kind of patient.

Author Response

Comment 1: Page 2, “Initially misdiagnosed.” What kind of misdiagnosis was it? What disease was diagnosed? Was the existence of the disease missed?

Response 1: I have clarified this in the Case Presentation section and it is highlighted in the text.

Comment 2: "HPB” should be fully spelled in their first appearance in the main text.

Response 2: I have spelled out HPB at its first appearance in the main text. It is highlighted at the end of the introduction section.

Comment 3:  Figure 1, authors should describe the imaging findings of the MRI scan in the legend. Also, authors can indicate where the problem, biliary injury, is with arrows in the image for readers to understand.

Response 3: I have provided a figure legend and modified the figure by adding an arrow pointing to the lesion for better clarity.

Comment 4:  Figure 2, the words are too small to see. Please create a better “Flowchart” for readers

Response 4: I have enlarged the image of Figure 2 for better visibility. If further enlargement is needed, it can be adjusted directly by the journal.

Comment 5: “gamma-GT” should be describe in fully-spelled

Response 5: I have spelled out at its first appearance and it is highlighted in the text.

Comment 6:  In Discussion, “Recent studies have shown that this type of biodegradable stent is not inferior to traditional plastic stents in maintaining biliary duct patency and has significantly reduced the number of reinterventions required for replacement and removal, thereby lowering overall treatment costs.” The reference #7 (Eur J Radiol. 2020;125:108899) should be inserted in the end of the first sentence where the content is described.

Response 6: I have inserted reference #7 (Eur J Radiol. 2020;125:108899) at the end of the first sentence in the Discussion section, where the content is described.

Comment 7: Table 2, which one was used for this patient? Why and how did authors choose the one? Please describe in Case Presentation or Discussion.

Response 7: I have specified which stent was used for this patient and explained the rationale behind the choice in the Discussion section. This information is now highlighted in the text.

Comment 8: In Conclusions, “in managing these patients.” Readers may not understand what  “these patients” are. Authors should describe here again what kind of patient.

Response 8: I have clarified in the Conclusions section that 'these patients' refers specifically to those with complex biliary tract injuries.

Round 2

Reviewer 2 Report

Comments and Suggestions for Authors

According to the comments from the Reviewers, the Authors responded adequately and conducted several modifications appropriately. This seems a quite well-written and reshaped manuscript.  Therefore, this can be suitable for publication in the journal.